# Indoor Vegetable Production: An Alternative Approach to Increasing Cultivation

**DOI:** 10.3390/plants11212843

**Published:** 2022-10-25

**Authors:** Peter A. Y. Ampim, Eric Obeng, Ernesto Olvera-Gonzalez

**Affiliations:** 1Nutrition and Human Ecology and Cooperative Agricultural Research Center, Department of Agriculture, College of Agriculture and Human Sciences, Prairie View A&M University, Prairie View, TX 77446, USA; 2Laboratorio de Iluminación Artificial, Tecnológico Nacional de México Campus Pabellón de Arteaga, Carretera a la Estación de Rincón Km1. 1, Pabellón de Arteaga, Aguascalientes 20670, Mexico

**Keywords:** urban farming systems and facilities, plant growth factors, population growth, climate change, urban crop production

## Abstract

As the world’s population is increasing exponentially, human diets have changed to less healthy foods resulting in detrimental health complications. Increasing vegetable intake by both rural and urban dwellers can help address this issue. However, these communities often face the challenge of limited vegetable supply and accessibility. More so, open field vegetable production cannot supply all the vegetable needs because biotic and abiotic stress factors often hinder production. Alternative approaches such as vegetable production in greenhouses, indoor farms, high tunnels, and screenhouses can help fill the gap in the supply chain. These alternative production methods provide opportunities to use less resources such as land space, pesticide, and water. They also make possible the control of production factors such as temperature, relative humidity, and carbon dioxide, as well as extension of the growing season. Some of these production systems also make the supply and distribution of nutrients to crops easier and more uniform to enhance crop growth and yield. This paper reviews these alternative vegetable production approaches which include hydroponics, aeroponics, aquaponics and soilless mixes to reveal the need for exploring them further to increase crop production. The paper also discusses facilities used, plant growth factors, current challenges including energy costs and prospects.

## 1. Introduction

Food production is very essential for the survival of humankind. Population projection by the medium-growth projection scenario of the United Nations indicates that the world population will reach 9.7 billion by 2050 [1,2]. Currently global consumption of food commodifies is increasing with population growth [3], hence agricultural production must continue to increase to meet this growing demand. However, agriculture is faced with several challenges including climate change and weather extremes, land degradation, expansion of drylands, shrinking supply of freshwater, urbanization, growing price tag of agribusiness including fertilizer, fuel, pesticides, and transportation costs [3]. Globally arable land per capita in 2050 is estimated to reduce by one-third the size documented in 1970 [4] indicating that less land will be available for agriculture. More so, one third of the human population is already estimated to be suffering from one or more forms of hunger or malnutrition [5]. This situation is likely to continue unless sustainable solutions are developed to address the current challenges facing food production and supply. Historically food production outpaced demand. For example, though the world’s population increased six- to seven-fold between 1800 and 2000, from less than one billion to six billion, world food production increased ten-fold within that same period [6]. However, the current projections indicate the level of food production success attained thus far may not continue 2050 and beyond. Hence, this calls for exploring alternative approaches such as indoor farming (IF) for growing food to meet the needs of our exponentially growing population. IF can facilitate the frequency of food production and the quality of food produced. This is because it provides growers the ability to create the desired conditions for crop growth regardless of outdoor weather conditions. For example, environmental factors affecting crop production such as temperature, relative humidity, carbon dioxide, and air circulation can be controlled in indoor farm environments. Similarly, challenges posed by soil factors in the open field including fertility, salinity, pathogens and pests as well as weeds and landscape challenges such as topography can be avoided in indoor farms as soil is not used as a medium. IF is often called other names in the crop production literature including controlled environment agriculture (CEA), protected environment agriculture, plant factories (especially in Asia), closed loop systems, urban agriculture, soilless growing and vertical farms [7]. However, in this paper it will be mostly referred to as IF or CEA.

Indoor farms represent a group of integrated technologies that can be configured in different ways for producing different crops [8]. These crop types include leafy vegetables, herbs, tomatoes, flowers [9] and microgreens [10]. The structures used for production of the crops include greenhouses, high tunnels, buildings, or containers with artificial light sources [11]. These structures can protect the plants from fluctuating abiotic and biotic stress factors affecting crop production. In addition, IF, uses less resources including water, pesticides, land, and soil. Increasing the attention given to indoor production is very vital because of the following reasons. About 25% of the world’s population suffer from micronutrient deficiencies and related health problems [12]. More so, the rapid growth of low-income urban dwellers coupled with widespread poverty, and dietary changes towards calorie dense and nutrient poor foods has increased the number of people suffering from micronutrient deficiencies. Though accessibility to affordable vegetables is a precondition for their consumption, a large share of poor and rich households in both high- and low-income countries do not have regular access to affordable and fresh vegetables [12]. People in urban areas mostly rely on the food supply chain for their vegetables. However, the supply chain may be poor or unreliable, seasonal, and production and price may be volatile, thereby making the fresh vegetables out of the physical or economic reach of millions of urban households [12]. Reducing the production and availability gap therefore will require increasing the production outlets including IF.

Outdoor vegetable production on the other hand in open fields exposes crops to abiotic (e.g., humidity, temperature, precipitation, wind, salinity) and biotic stress factors (pests, insects, soil microbes, weeds) which can make outdoor production seasonal and sometimes very volatile. Climate change and its associated unpredictable weather patterns and extremes add to these uncertainties [12]. Some of these factors can be difficult to predict further revealing the need for exploring more efficient and timely alternative means of increasing food production. In addition, the perceived risks associated with pesticide use on vegetable crops can be curtailed through indoor production. Additionally, water use in open field production systems is about 95% greater than in IF [12]. Reduction in water use is very critical in this era where the world is faced with increasing water scarcity, depletion of ground water and unsustainable management of surface water.

The use of fossil fuels in conventional farming to power tractors and other farming implements contributes to air pollution, greenhouse gas emission, global warming, and health complications for humans [13]. In the U.S. and most industrialized countries, farming activities including plowing, planting, fertilizer application, pesticide application and transportation account for about 20% of the total fossil fuel used [14,15]. IF in or near cities can cut down on fossil fuel needs for food production thereby reducing pollution [16].

IF also saves land space compared to conventional farming because both horizontal and vertical spaces are often used more effectively leading to maximization of the units of vegetables produced per acre [15,17,18]. Adoption of IF can reduce agricultural land use up to 20 times and hence can lead to a 1/20 ratio improvement in land use [15]. Adding one or two additional floors to a vertical indoor farm will result in 1/40 and 1/60 ratio improvements in land use, respectively. As such in big and congested cities where land availability is a challenge and very expensive, IF can be used to effectively produce food.

IF also permits more production cycles and therefore food production frequency. Since the crops are protected from external environmental factors, a variety of crops can be grown year-round using IF [19,20]. In general, year-round production of crops using conventional farming approaches is more challenging if not impractical in some cases because of seasonal changes and differences in crop adaptations. Cicekli and Barlas [21] reported that 23 times more lettuce can be produced in a vertical indoor farm compared to the same land space in conventional farms. Besides, consumer interest in local foods is increasing and as a result, the market share of IF is rapidly growing in the produce industry because of its ability to meet the unmet demands for produce grown locally [7]. IF also reduces produce contamination and is reported to increase shelf life by two to three weeks [8].

IF spans several subject areas as such it is difficult to provide a completely exhaustive review covering all its relevant aspects. As a result, published reviews on IF turn to focus on specific topics such as greenhouse types and their structural components [22], components of soilless mixes and potting mixes [23], energy efficiency [24], IF applications in various environments including tropical and extreme [25,26], light technologies [27], hydroponic technologies [28], aeroponics [29], aquaponics [30]. Other reviews and reports have provided broader reviews with varied foci such as the history, industry landscape and impacts and discussions on some of the structures and production systems [10,28,31,32,33,34,35].

Therefore, the purpose of this paper is to provide a more comprehensive review on IF that includes the facilities and growing systems/techniques employed in IF as well as its economics, current challenges, and prospects. This review also highlights vegetables and herbs studied for indoor production, plant growth factors affecting indoor crop production and the impacts of plant nutrition on produce quality. A review of this nature that covers more ground will serve as a useful reference resource for researchers and other interest groups interested in exploring IF. 

## 2. Facilities Used for Indoor Vegetable Production

Common facilities used commercially for IF include glass- or poly- greenhouses, vertical farms, low tech plastic high tunnels or hoop houses, containers and indoor DWC [10] and *s*creenhouses [36]. 

### 2.1. Greenhouse

A greenhouse is a structure covered with a glass or plastic roof and side walls with full climate control [36]. They are often designed in different configurations and can be free-standing or gutter-connected [36] and can have different levels of sophistication [22,36]. Greenhouses are typically constructed with galvanized steel tubing in the U.S. [36] but can be built from other materials like wood in countries like India [22]. Greenhouse types are often categorized based on utility, shape, and the construction material used [22]. Modern greenhouses use lighting, but it is supplemental to natural sunlight [8]. As result, hydroponic production is a greenhouse environment can be cheaper because sunlight is the main source of energy for crop production. Supplementing sunlight with LED lighting has been reported to have a potential energy savings of 70% when used in a 70:30% ratio (i.e., 70% sunlight:30% LED lighting) [37].

Greenhouse vegetable production is an intensive system that can extend the growing season [38,39]. Greenhouse production has been highly developed since the 1970s, in many western and Asian countries including the Netherlands, Israel, the United States, and Japan [27,40,41,42]. Greenhouses have improved in the areas of climate control, advanced planting, use of environmentally friendly materials, and effective management practices. An advanced greenhouse vegetable production system has several characteristics. First, they are scientifically designed to increase light exposure, save cost, and reduce environmental impact through the use of biodegradable plastic films which can decompose if released into the environment and generally have a lower carbon footprint in terms of manufacturing and decomposition [39,40,43]. Secondly, environmental factors such as temperature, CO_2_, humidity, and light are controlled by mechanized system for seeding, planning, harvesting, packaging, and transportation [44]. For example, through the use of robots, modern greenhouses can autonomously measure the microclimate [45], carry out spraying [46] and harvesting [47,48,49].

Greenhouses are used to produce high value vegetables that are difficult to produce outdoors such as colored pepper, cucumbers, beefsteak, and other tomatoes [50,51]. In the United States, greenhouses are popular in vegetable producing areas such as California and Florida. Despite these advantages, greenhouses can be costly to build and operate. Leafy vegetable crops commonly grown indoors include arugula (*Eruca sativa* L.), broccoli (*Brassica oleracea* var. *italica*), cabbage (*Brassica oleracea* var. *capitata*), lettuce (*Lactuca sativa*), mustard (*Brassica juncea* L.), and a host of others. A list of some of these vegetables is summarized in Table 1 while the types of facilities used to research their production indoors are summarized in Table 2.

### 2.2. High Tunnels or Hoop Houses

High tunnels look like traditional plastic-covered greenhouses, but they use a completely different technology. High tunnels are usually constructed with a pipe or galvanized tubing covered with a single layer of greenhouse-grade 4- to 6-mil plastic and do not have electrical service, automated ventilation, or heating system [78,79,80]. They are usually constructed on bare ground and depend mainly on passive solar heating and passive ventilation [81]. Though temperature in high tunnels is usually a couple of degrees higher than the temperature outside, it is necessary to have a standby portable heater or other methods of heating to protect crops against unexpected low temperatures in the spring or fall. Water supply to high tunnels is mostly through drip irrigation but can also be done using small sprinklers or hand watering [81]. High tunnels are also designed to have manually operated or automated rollup sides, as well as end and ridge vents for ventilation. While planting can be done directly on the ground in a high tunnel, it can also be done on raised beds or containers [81].

The framework and construction of high tunnel varies from one country to another. In South Korea, single-bay high tunnels covered with single- or double-layer plastic are erected in the field during the growing season, and then removed and stored at the edge of the field at the end of the season [79]. In Spain, high tunnels have flat, sloped roofs to allow rainwater to runoff. The structure has a wire framework that runs between metal or wood posts and is covered with a two-layer plastic film. In tropical regions such as Taiwan, Thailand, the Philippines, and Indonesia, high tunnels are made from pipe frames, and covered first with a screen material, followed by a plastic film on top. The plastic film can be removed to expose the screen material which facilitates ventilation. In India however, high tunnels are built of sturdy bamboo frames, and covered at the top with a single layer of plastic film, and on the sides by jute, which facilitates ventilation while excluding insects [79]. In general, they are quonset or gothic shaped and can be designed as more permanent or mobile structures.

### 2.3. Screenhouse

A screenhouse is an affordable intermediate technology between open field and greenhouse cultivation. This structure is usually built with metal columns supported by cables, with roof and side walls installed using porous screens [36]. Commercially available screens come in different colors, material types, and porosity, and these characteristics affect their optical and aerodynamic properties [36]. The screenhouse characteristics can be modified to improve microclimate conditions generated inside [82,83].

Screenhouses or screen covered structures can be used to produce summer crops when production in plastic greenhouses ceases due to high temperatures. Screenhouses have the ability to reduce the intensity of incoming solar radiation [84]. In general, screenhouses are used for agricultural applications such as reducing solar radiation, modification of the solar spectrum (colored net), modification of the micro-environment of a crop, protection from insects and birds, protection from hail, heavy rainfall, snow, and wind [84]. A misting system can be added to screenhouses to reduce ambient vapor pressure deficit (VPD), thus improving growing conditions [84]. Disadvantages of screenhouse include the lack of protection against rain and extremely low humidity [85].

### 2.4. Indoor Vertical Farms/Gardens 

Indoor farms have existed in Asia for quite some time and are known as Asian indoor farms [86]. The environmental conditions in these farms such as carbon dioxide, light, relative humidity, and temperature are managed by computer and sensor controls [86]. In Japan, a lot of these viable indoor farms are in peri-domestic areas [87]. The main challenge for these enterprises in Japan is the increasing land cost, which has made producers reassess their economic viability. Indoor farms operate in controlled environments and therefore require less water than outdoor farms, hence they have high probability to be adopted in desert and water limited environments such as Middle East and Africa, and in small and urbanized countries such as Israel, Japan, and the Netherlands [87].

Vertical farming is an indoor growing technique in which crops are grown in vertically stacked tiers or on vertical surfaces [8]. The crops are grown mainly with artificial light from light emitting diodes (LEDs) in structures such as traditional warehouses, previous industrial spaces such as old mill buildings or shipping containers [8]. Vertical farms are very complex and allow for more cultivation area on a relatively small base area [86]. Introduction of the vertical farm concept was aimed at increasing agricultural land by building upwards, that is increasing the effective arable area on the same footprint of land [87,88]. The main advantages are that vegetables can be produced in close proximity to consumers and the production in controlled environment allows for higher yields [86,87]. Vertical farming is also attractive in parts of countries such as China that suffer from environmental pollution and soil depletion but have a high demand for clean, green, and gourmet (CGG) foods.

Crop production in indoor farms can take place all year round since the system does not depend on soil and climatic factors. Some of the other issues with outdoor production such as soil erosion are avoided. Producers also have a better control over the use of pesticides and fertilizer. The greatest advantage of vertical farms is independence from climatic conditions. Hence, healthy crops can be grown in a sustainable way even in cities or locations with contaminated soils or weather extremes [86,87]. Despite the advantages, vertical gardening has its own challenges, which include high energy cost, high technical expertise needs, and high cost of maintenance [89]. 

## 3. Growing Systems/Techniques Used in Indoor Farming

Several growing systems or techniques are employed for growing crops in indoor farms. These include soilless mixes, hydroponics including aeroponics and aquaponics [10].

### 3.1. Soilless Mixes

Growing plants require growing media to hold them in position and to supply them with nutrients needed for growth. Straight soil from the garden may not be ideal for container grown plants since they may contain too much clay, which holds too much water when wet and reduces the supply of air to the plant [90]. A container medium must therefore be porous to promote air and water movement. Examples of soilless growing media available to growers include peat-lite mix, peat moss, perlite, coconut coir dust, jute and kenaf fibers, rock wool, and vermiculite (Table 3). Soilless mixes are generally sterile and contains few nutrients, hence fertilizers must be applied to supply nutrients to the growing plant. Garden soil can be mixed with soilless media to increase weight and water holding capacity.

### 3.2. Hydroponics

Hydroponics is a system whereby the plants being grown are supplied with water and nutrients, in the absence of soil. Thus, the components of a hydroponic system include a plant culture and nutrient supply (fertigation) units, water equipment including aerators and nutrients needed by the plants grown [33]. In hydroponics, the growing plants are held in net pots or on clay pellets, perlite, rock wool which are chemically inert [9]. Plant culture systems in hydroponics are open or closed systems with the open systems being most common even though closed systems are more environmentally friendly [33,93]. In the closed loop systems water is recycled but in the open system the nutrient solution delivered to the plants roots is drained after a single application and not reused [33,93]. The plant culture system can also be a liquid or aggregate system. In a liquid system, plants roots hang in a nutrient solution or are misted whereas in the aggregate system, roots are grown in inert media and are irrigated with a nutrient solution [33,93]. If a hydroponic system uses a wick and growth media with very high capillary action to draw water to plant roots it considered a passive system. On the contrary, it is considered an active system if nutrient solution is actively passed over plant roots with the help of pumps [93,94]. This implies that the plant roots in these systems can either be fed with a flow of solution or suspended in solution full time [95]. 

The different types of hydroponic systems used often include (1) aeroponics (liquid and closed or open), (2) drip system (aggregate and closed or open) (3) Nutrient film technique, (4) deep water culture (liquid and closed), (5) ebb and flow (aggregate and closed) and (6) wick system (7) aquaponics (liquid and closed or open) [33]. It must be noted that usually the cost of these systems varies by design, operational characteristics, and reliability [96].

In aeroponics plants are grown in the air with water and nutrients sprayed onto their roots at regular intervals (Figure 1a). The plants are suspended in air with support from boards, foam sheets or other methods including baskets on top of a closed cylinder or trough [9,94]. As result of the suspension arrangement of the plant roots, aeration around them is at maximum and this promotes faster plant growth [94]. Aeroponic systems require precision sensing technology to deliver the nutrient doses in a timely and effective manner to maximize outcomes. They provide the highest nutrient use efficiency because plants are only provided what they need, and unused nutrients are recycled. However, for the systems to work effectively, the root environment must be maintained at 100% relative humidity to prevent root dehydration [94]. Though aeroponics systems can produce yields 10 times greater than soil production, pump malfunction and/or electricity failure can be detrimental to root health including desiccation and death, and ultimately the killing of plants [94].

In the drip system (Figure 1b), nutrient solution is supplied directly to the base of plants in a regulated manner through tubes and drip emitters. In closed systems, the nutrient solution is delivered at set time intervals and the excess solution is returned into a reservoir (Figure 1b). In non-circulating systems, however, nutrients are supplied at a consistent rate through slow dripping. This system can be designed to grow different varieties of plants [98]. 

The nutrient film technique (NFT) (Figure 1c) is considered the most scalable and is often operated as a closed system. In this system, plant roots suspend in a trough or tube inclined at a slope of about 1–2% and nutrients are supplied to the plants through a continuous flow of the nutrient solution through the trough from the top to the lower end by gravity [33]. Plant spacing in the trough is done according to recommendations for the crop type and plants are secured in place using foam net pot inserts [96,98].

The deep water culture (DWC) system (Figure 1d) is designed to grow plants in boards floating in nutrient solution. The roots are often aerated directly using air stones or diffusers (Figure 1d). Hence in this system, the plant roots are immersed while the foliage stays above the nutrient solution. It is also suitable for a variety of plants especially those with large rooting systems. It is called other names including deep flow technique, floating raft technology and the floating root system [33,98].

In the ebb and flow system (Figure 1e), plant growing in an inert medium are flooded periodically with nutrient rich solution using pumps and drained back to a reservoir below the growing bed by gravity for reuse (Figure 1e). This system also called flood and drain system can be set up with automatic drains allowing the system to flood and drain faster. It is advisable not to cultivate plants that grow too large in this system because of the limited size of the growing bed [96,98]. 

The wick system (Figure 1f) is the simplest of hydroponic systems and is completely passive. In this system plants are grown in an aggregate medium supplied with a wick which runs into a nutrient solution reservoir. The wick absorbs the nutrient rich solution and moves it into the medium in which the plants are growing (Figure 1f). It is most suitable for growing small plants and herbs at home and not for commercial purposes [33,98]. Commonly grown crops in hydroponic systems include herbs or microgreens such as basil, watercress, dill, oregano, bok choi, leafy greens such as lettuce, kale, spinach and vine crops such as tomatoes, cucumbers and peppers [33].

Aquaponics is a system which integrates aquaculture (fish production) with hydroponics (soilless cultivation of plants) in a single production system [99,100]. In aquaponics, the fish are fed, and their waste is converted into nutrients for the plants. The system relies on proper functioning of microbes since they are key in the efficient conversion of fish waste into nutrients for the plants [9]. A successful aquaponics system requires knowledge in aquaculture, hydroponics, and maintaining microbes and nutrients. Aquaponics is considered as subset of integrated agri-aquaculture systems (IAAS) [101] because while IAAS uses several separate aquatic animal and plant production technologies in various settings, aquaponics is closely linked with combining tank-based fish culture technologies for example recirculating aquaculture systems (RAS) with aquatic or hydroponic plant culture technologies [102]. Therefore, the key elements of an aquaponics system are the fish tank, the water filtration system, water flow and aeration, type of fish and plants. The other vital component is the plant culture system which can be DWC, ebb and flow or NFT (Figure 2) depending on the type of plant selected for cultivation [103]. The DWC is used mostly for fast-growing and early maturing plants often harvested in whole (Figure 2a) while the ebb and flow system lends itself to growing the highest diversity of plants (Figure 2b). For instance, the ebb and flow can accommodate growing early maturing lettuce next to tomatoes which takes a longer time to mature [103]. NFT aquaponics systems also provide flexibility for cultivating plants with short and long growth cycles (Figure 2c) [103]. The importance and considerations for choosing the appropriate components for aquaponics systems are summarized in Table 4. 

### 3.3. Advantages and Disadvantages of Hydroponics

In general hydroponics have numerous merits but also have demerits. These attributes have been articulated by several authors [28,93,96,99,100,104,105]. The advantages include (i) limited space requirement and therefore permits widespread use in many places including in space, (ii) elimination of seasonality and increased frequency of production as crop cycles can be repeated as many times as possible, (iii) eliminates the need for cultural practices like crop rotation and weed control required in open field and soil-based production systems, (iv) offers opportunities for infusing ergonomics into certain aspects of leafy vegetable crop production such as raising lettuce, kale, mustard, spinach, endive, Swiss chard and many Asian greens and small fruits like strawberries off the ground to a desired height to ease cultivation and harvesting activities, (iv) reduce transplant shock and provide 30% to 50% faster plant growth rates compared to soil-based crop production methods, (v) offer enhanced nutrient use efficiency permitting at least 20% higher yields compared to soil-based cultures, (vi) the absence of soil in this systems, the recycling of the nutrient solution coupled with their operation in a controlled environment helps minimize the incidence of plant diseases and pests and hence the regular need for use of pesticides, (vii) hydroponically grown plants use considerably less water (80–95%) than those grown in the field because of less water loss through evaporation providing savings on water use in crop production.

Shortcomings of hydroponic systems include (1) higher initial and operational costs compared to soil culture however it must be noted that usually the cost of hydroponic systems varies by design, operational characteristics and reliability [96], (2) proper operation of the systems involves skills and knowledge, (3) there is a potential for easy spread of fungal pathogens like fusarium and verticillium (4) requires more intensive management, (5) since the movement of water in some hydroponic systems is power driven, electricity outages could hamper the function of these system since there will be no water flow in the absence of power. The electricity requirement can also lead to high energy costs. Table 5 summaries the pros and cons of the individual hydroponic systems.

The advantages and disadvantages of aquaponics are akin to those of hydroponics. In general, they are amenable to small- and large-scale production operations and can be established in many locations including on non-arable lands [100]. They also do not require weeding, tillage and other cultural activities required for soil cultivation. Water use is also lower, and nutrients are not wasted [100]. The disadvantages of aquaponics include high initial costs compared to soil production and hydroponics, knowledge requirement on fish, bacteria, plant production, limited management options, different requirements for both fish and plants, limited room for mistakes, dependability on electricity supply and high level of energy consumption including a requirement for a certain optimal temperature range [100]. The strengths and weaknesses of the major types of aquaponic systems namely DWC, NFT and ebb and flow have been extensively covered by [100] therefore they will not be discussed here.

The various hydroponic and aquaponic systems discussed above can be designed to suite household gardening, research- and large-scale commercial production needs [93,94,105,106]. The global market for hydroponic systems which provides an idea of their extents of use is expected to reach USD 25.1 billion by 2027 [107]. The global market for aeroponics alone is projected to hit USD 3.53 billion in four years (i.e., by 2026) [108]. Table 6 below presents a summary of the scales of use of the various systems.

## 4. Plant Growth Factors Affecting Indoor Production

### 4.1. Light Quality and Photoperiod

Research on indoor production has focused on factors such as light quality, light intensity, and photoperiod (Table 7). Light is very vital for indoor production because of its impact on plant growth and yield. Light affects plant growth rate including stem thickness, branching, and rooting through its role in photosynthesis, and developmental processes such as seed germination and flowering [109,110]. Studies show that applying supplemental lights of various qualities (UV-A, blue, green, red, far-red, and white LED) had significant effect on phytochemical contents (anthocyanins, carotenoids, chlorophylls and flavonoids) of lettuce leaves [111,112]. Carotenoid biosynthesis in fruits and vegetables is highly dependent on light intensity and quality [113]. In greenhouses the absence of UV-B radiation can affect flavonoid content in leafy vegetables with direct effect on flavor and appearance [114,115].

Modern indoor production relies on light emitting diodes (LEDs) to supply light for plant growth because traditional light sources have a spectral mismatch between the emitted spectrum of the lighting and plants [110]. LED lights are more attractive for indoor production due to less heat production compared to incandescent light, and they do not require ballasts. Basically, LEDs are specialized diodes than can pass current in forward direction but block current flow in the reverse direction [9]. As the LED light technology is still improving, growers and researchers are experimenting with different spectral composition and crop varieties. LED artificial irradiation is an essential factor in IF. An important feature of LED technology is that it permits the setting of different wavelengths, called light recipes (light emission produced by a combination of wavelengths). These light recipes can improve the development and yield of crops from sprouting to flowering, enhance plant elongation, and increase edible biomass production. Additionally, it contributes to the generation of higher nutritional content like antioxidants (phenolic acids, flavonoids, anthocyanins), vitamins (vitamin K, A, C, and B), minerals (Mg, Fe, K, Zn, and P, among others), proteins, fibers, sugar, and others [116,117,118]. LEDs are versatile and can be digitally programmed to turn on and off without waiting for a lamp to start and for run-up time for cooling before restarting, which is characteristic of some traditional lighting sources [119].

Photosynthesis is driven mainly by red and blue light consequently providing the right doses of these lights can efficiently promote plant growth [27,60]. Other lights that are not photosynthetically efficient may convey environmental information to the developing plant. For example, while far-red light affects phytochromes and leads to changes in gene expression, plant architecture, and reproductive responses, green light opposes the effect of red and blue light [111,120,121,122]. Red light induces plant physiological responses such as leaf development, stem elongation, root to shoot ratio, and chlorophyll and carbohydrate accumulation [123,124,125,126,127]. The energy of blue light is often significantly lost because of absorption by accessory photosynthetic and non-photosynthetic pigments such as anthocyanin that are inefficient in energy transfer to chlorophyll [121]. However, blue light influences photosynthetic activity by inducing the opening of the stomata [128]. It also affects chloroplast movement within the cell and increases the number of stomata and leaf thickness [123,125] in the short and long term respectively.

Photoperiod is known to regulate flowering in plants [128,129,130], hence in indoor production systems, photoperiod can be modulated to promote flowering in ornamental crops [131]. Notwithstanding, photoperiod has not been fully explored in indoor cultivation of leafy greens. This may be due to the reason that photoperiods (14–18 h light) routinely used have not had pronounced effects on growth and phytonutrient accumulation compared to light quality and quantity [131,132,133]. Majority of studies with LEDs have been conducted using photosynthetic photon flux density (PPFD) of between 150 and 300 μmol m^−2^ s^−1^ that is 7.5–15% of full sun [9]. At the upper limits within this range, leafy greens have responded with higher growth rate, as it provides more photons to drive photosynthesis which results in increased biomass production.

### 4.2. Carbon Dioxide

In indoor production systems, carbon dioxide (CO_2_) concentration of 1000–1200 μmol mol^−1^ is usually maintained to increase yield and early maturation [115,134]. Studies on CO_2_ enrichment in greenhouse settings show that yield of fruits and vegetables such as lettuce increased up to 30%, with accompanying significant increase in chemical composition. A higher yield and content of flavonoids and caffeic acid was realized in two red lettuce cultivars when CO_2_ enrichment was applied up to 1000 μmol mol^−1^, and this could be attributed to the increase in sugars which serve as precursor for flavonoids biosynthesis [135]. Red lettuce cultivars seem more responsive to high CO_2_ than green lettuce, which was indicative of higher content of secondary metabolites and antioxidant activity [136]. Lettuce plants inoculated with arbuscular mycorrhizal fungi showed a negative effect with elevated CO_2_, as photosynthates were consumed antagonistically for shoot growth and mycorrhizal colonization instead of being used for leaf biosynthesis [137]. Elevated CO_2_ increased the amount of carbohydrates in tomatoes grown in a greenhouse, whereas crude protein, vitamin C, organic acids, fat, and ash decreased [138]. Also, mineral composition in the study was affected by genotype. Contrarily, a greenhouse study by Jin et al. [139] reports that CO_2_ enhancement through crop compost and animal manure increased sugars and ascorbic acid significantly, and decreased nitrate content in the leaves of celery (*Apium graveolens* L.), leaf lettuce (*Lactuca virosa* L.), stem lettuce (*Lactuca sativa* L.), oily sowthistle (*Sonchus oleraceus* L.), and Chinese cabbage (*Brassica chinensis* L.).

### 4.3. Relative Humidity

Humidity of air is expressed as either relative humidity (RH,%) or vapor pressure deficit (VPD, kPa) [115] and is an important factor that affects the growth of plants. Plants are damaged when they experience a high RH or low VPD due to reduced evapotranspiration rate and reduction in sap flow through the phloem. These reduce the translocation of ions inside plant tissues and result in nutrient deficiency [115]. This may not be an issue in indoor farms where relative humidity is well-controlled to improve crop yield and quality traits. For example, the use of a fogging system in a screenhouse to increase relative humidity has been reported to increase productivity and quality parameters of cherry tomato including lycopene content, vitamin C and antioxidant contents [115,140]. Leonardi et al. [141] and Rosales et al. [142] have reported similar effect of low VPD on the lycopene content of salad and cherry tomato grown in greenhouses. High relative humidity (i.e., above 95%) at night can reduce tipburn on the leaves of butterhead lettuce because root pressure increases the translocation of Ca to leaf margins and helps alleviate Ca deficiency symptoms such as tipburn [115,143]. Hence, regulating the relative humidity in indoor production systems can maintain the quality of fruits and leafy vegetables and simultaneously affect their phytochemical composition and antioxidant capacity.

High VPD usually occurs simultaneously with high temperatures and solar radiation, which together induce oxidation stress and negatively affect the marketability of produce as well as carotenoid and mineral contents, but may increase antioxidant activity, phytonutrients, and sugars content [142,144,145]. A greenhouse study by Xu et al. [119] suggests that high VPD can increase chlorophyll content, chlorophyll a/b ratio and the soluble protein content in tomato plants. The study also indicated that high VPD conditions increased the enzymatic activity of Rubisco (ribulose-1,5-bisphosphate carboxylase/oxygenase) and increased photosynthetic capacity of plants compared to low VPD conditions. Similarly, tomato plants grown under high VPD conditions had higher fruit yield and better quality (i.e., texture, color, and sugar content) compared to those grown under low VPD [146].

### 4.4. Temperature

Ambient air temperature regulation in indoor production systems has significant effect on vegetable quality. Schonhof et al. [147] reported that the combination of elevated air temperature and low radiation may have a positive effect on ascorbic acid, glucosinolates, and lutein content of greenhouse broccoli. Optimum plant growth and yield occur within species specific temperature ranges, hence low or high temperatures outside this range can hamper plant growth, plant development, yield and quality due to nutrient and hormonal imbalances, protein misfolding and reduced radical scavenging activity [148,149,150,151]. According to a study by Max et al. [152], fan and pad cooling in greenhouses covered with polyethene decreased the occurrence of blossom end rot and undersized fruits, and increased calcium content, whereas net covered greenhouses with mechanical ventilation decreased the occurrence of fruit cracking. Cumulative temperature (i.e., sum of daily temperatures above a specific temperature threshold during the growing period) within a greenhouse have been strongly correlated with the quality traits of tomato fruits such as firmness, electrical conductivity of fruit juice, soluble solids content (SSC), and content of phenolic compounds, while dry matter, ascorbic acid content, titratable acidity (TA) and pH of juice did not correlate with cumulative temperature [153]. Moreover, temperatures below 12 °C and above 32 °C can inhibit lycopene biosynthesis in tomato fruits [154], whereas day/night temperature differential is vital for carotenoid synthesis, especially lycopene [128]. Notwithstanding, Gautier et al. [155] reported that temperature between 21 and 26 °C decreased the total content of carotenes but not lycopene, while high temperatures (27–32 °C) had a negative effect on the content of lycopene and its precursors (i.e., phytoene, phytofluene, and neurosporene).

Root-zone temperature is equally important for vegetable growth and quality under indoor production systems. Findings by Urrestarazu et al. [156], suggests that heating the nutrient solution in soilless cultivation systems, may result in early maturation of melon fruit, increased yield, and SSC (°Brix) content, which is an essential quality trait for marketability of the final product. Furthermore, Cabañero et al. [157], reported that higher calcium uptake by pepper occurred at root zone temperatures of 35 °C, which mitigated negative salt effects and incidence of blossom end rot. Contrarily, root-zone temperatures above 20 °C negatively affected tomato fruit quality but had a beneficial effect on fast growing vegetables [158]. Studies by Kafkafi [159] and Tindall et al. [160] suggest that 12 to 20 °C temperature of nutrient solution increased water flow by 250% through the stems of tomato plant, which subsequently resulted in higher nutrient uptake and better fruit quality. Indirect effect of temperature on vegetable quality have also been reported. A study by Elad et al. [161] showed that passive heating decreased the incidence of *Sclerotinia sclerotiorum* and *Botrytis cinerea* on sweet basil plants under controlled environments. They suggested that this observation was probably due to higher plant resistance to the pathogens rather than direct lethal effect on the pathogen.

It must also be noted that there is a relationship between temperature and oxygen as the solubility of oxygen in water depends on temperature [162]. Al-Rawahy et al. [163] noted that high temperate of nutrient solution is a stress around the rootzone and is an important limiting factor for hydroponic crop growth because of its impacts on dissolved oxygen (DO) levels in the nutrient solution. Al-Rawahy et al. [163] studied the effect of nutrient solution temperature on its oxygen level, and the growth, yield and quality of cucumber (*Cucumis sativus* L.) grown hydroponically in a greenhouse and found that cooler nutrient solutions positively influenced DO levels in the nutrient solution in both the feeding tank and drains. This was accompanied by higher levels of oxygen use in the rootzones of the cooled treatments compared to the controls with no cooled rootzones. The study also reported that cooled rootzone temperature had a positive impact on the growth, production, and quality attributes of cucumber.

## 5. Impacts of Plant Nutrition on Produce Quality

Availability of essential nutrients in soil or soilless culture in indoor production systems is an important crop management practice [115,164]. In indoor production, the vegetables grow in various media such as peat moss, coconut husk, and fiber mats (Table 3), need nutrient amendments. Adequate supply of primary macronutrients such as nitrogen (N), phosphorus (P), and potassium (K) as well as secondary macronutrients like calcium (Ca), magnesium (Mg) and sulfur (S), and micronutrients including iron (Fe) and silicon (Si) can improve vegetable yield and produce quality. In a greenhouse pepper study, Flores et al. [165] reported significant increase in the concentration of β-carotene, lycopene and lipophylic antioxidant compounds content with increasing N application. However, in a study by Yasuor et al. [166], pepper fruit quality (β-carotene and lycopene content and antioxidant activity) was not affected by N concentrations at 9.2, 56.2, 102.3, and 158.5 mg L^−1^. Excessive application of N fertilizer on the other hand can lead to changes in nutritional and commercial quality traits of vegetables such as decrease in the concentration of soluble sugars, carotenoids, and vitamin C, while increasing the concentration of nitrates, TA, and acid: sugar ratio [167,168,169,170].

Nitrate accumulation in leafy vegetables can be reduced by proper management of N in soilless culture [171]. This is important because nitrate can be converted to nitrite post-harvest and become harmful to consumers because it can cause methaemoglobinaemia or carcinogenic nitrosamines [172]. Three strategies have been proposed in scientific literature for reducing nitrate accumulation. These include: (i) replacing nitrate-based fertilizer (e.g., calcium nitrate) with chloride-based fertilizer (e.g., calcium chloride), (ii) depriving the leafy vegetable of nitrate several days before harvesting, for example 2–15 days depending on species, or (iii) partially replacing nitrate with ammonium nitrogen [171,173,174,175]. Borgognone et al. [173] reported that replacing nitrate with 80% chloride (20:80 NO_3_^−^:Cl^−^ ratio), increased antioxidant activity, flavonoids, and total phenols in cardoon leaf extract without detrimental effect on productivity. Nitrate levels in some leafy vegetables have been lowered when nitrate was withdrawn from solution few days before harvesting (i.e., 5–15 days) for cardoon [173], and 2–5 days before harvesting for chicory, lambs’s lettuce and rocket [174,176]. In the study by Borgognone et al. [173], they also found a linear increase in the total phenols and flavonoids in the leaf tissue of cardoon after 5, 10, or 15 days of nitrate withdrawal. The synthesis and accumulation of flavonoids under N deficiency is possibly due to the activation of key enzymes involved in the flavonoids pathway and the involvement of flavonoids in the reactive oxygen species (ROS) scavenging cascade [177,178]. Nitrogen source can also affect vegetable quality. For example, Zhang et al. [179], realized a decreased nitrate concentration from 79 to 89% when spinach was fertilized with ammonium compared to nitrate nitrogen. Moreover, tomato plants had better flavor when 10% of the total N supplied was in ammonium form [180,181], and this could be due to elevated glutamine and glutamate levels.

Potassium plays important roles in many plant cell processes such as carotenoid biosynthesis, through its action on key enzymes such as phosphofructokinase, pyruvate kinase as well as on precursors of pyruvate and glyceraldehyde 3-phosphate [115,182]. Trudel and Ozbun [183] reported that carotenoids and lycopene content increased by 40% when K was increased from 0 to 8 mML^−1^ K in nutrient solution. When Taber et al. [184] conducted a greenhouse study to investigate the effect of four K fertilizer rates (i.e., 0, 2.5, 5.0 and 10 mML^−1^) on lycopene of three tomato cultivars, they found that increasing fertilizer rate from 0 to 10 mML^−1^ increased K concentration by two-fold, and increased fruit lycopene as well as its colorless precursors (phytoene and phytofluence) depending on the genotype. Likewise, Almeselmani et al. [185] reported an increase in the quality traits (SSC, ascorbic acid and lycopene content) of tomato grown in a greenhouse as K concentration in nutrient solution increased up to 300 mg L^−1^ (7.7 mML^−1^). Fanasca et al. [186] realized a high lycopene concentration during the red and intense red stages of tomato fruit, when they received a high K concentration in nutrient solution. However, excess application of K should be avoided since it may lead to incidence of blossom-end rot and other physiological disorders.

## 6. Economics of Indoor Vegetable Production

The cost of new innovations and technologies are generally high, and this applies to setting up indoor production system. New technologies typically also have a high risk of failure, but cost goes down over time when the technology is perfected. One of the main questions is for example, can commercial scale IF compete with conventional production systems such as open field? Venture capitalists who are early adopters and high-risk takers offer real opportunities for economically viable production units. Their failures provide information for innovators to improve their products. They can also reap the early benefits of IF systems if their technologies are successful. Most IF systems rely on artificial lightning system for plant growth. As a result, 30% of the total energy cost can be tied to the energy cost for running these lights in the system while the remaining cost goes into climate control and systems operation [9]. A way to cut down on this cost is by using energy efficient LEDs. Some authors [33] project that an efficiency level of 50–60% in LEDs is necessary for LEDs to become cost effective for growing a diverse group of crops. Production of LEDs with 68% efficiency by Philips and the potential for a water-cooled LED light with the capability of being combined with a heat recovery system in the works in Finland are all necessary innovations to cutting energy cost in IF [167]. The market of LEDs is also expected to grow from USD 1.13 billion in 2018 to USD 6.78 billion by 2026, with a compound annual growth rate (CAGR) of 24.9% [187]. Increasing demand coupled with research and the associated economy of scale will reduce the cost of LEDs over time [27].

Another important factor is full costing which includes the social and environmental costs. Generally, the environmental costs for indoor farming are less than conventional open field farming, particularly when based on renewable energy sources. For example, the relative impact on water, land and biodiversity of indoor production is less than outdoor production in terms of gas emission, pesticide use, and fertilizer use efficiency. Other cost of open field farming including the cost to society in terms of environmental degradation is not included in the farmer’s cost. Nonetheless, even a full cost competitive framework which includes environmental and health externalities in the production costs, have found energy and capital costs to be the key cost factors in IF [12]. Some of the factors that can cause a tremendous shift in favor of IF crops are as follows: improvement in the efficiency and reduction in the unit cost of renewable energy, including dramatic cost reduction, efficiency in enhancing changes in lighting and solar power; low capital cost; availability of inexpensive and outdated factory buildings for remodeling; and government provision of subsidies on vegetables produced indoor [188]. Profitability of indoor farming however seems to vary with the facility type, growing system and the type of crop grown. Profits reported on a dollars per square foot basis was higher for indoor vertical and indoor DWC facilities ($14.88) compared to greenhouse facilities ($7.29). Similarly, leafy greens produced from hydroponic growing systems had higher profits than those produced across different growing systems (i.e., $17/ft^2^ vs. $13.83/ft^2^) [10]. Based on the crop grown, 40% profit margins have been reported for both leafy greens and microgreens compared to 30% and 10% for flowers and tomatoes, respectively [10].

Based on the environmental dimension (i.e., social and economic factors not included) alone however a study suggests that for now, conventional production appears to be more environmentally sustainable than IF even though the study acknowledges that this situation may change over time since IF technologies are becoming more efficient. This conclusion was reached when life cycle assessment was done for 1 kg of lettuce grown in a conventional production system in California and transported to a grocery store in St. Louis, Missouri compared to the same quantity produced from hydroponic and aquaponic systems in controlled environments of a greenhouse and vertical indoor farm. This study also concluded that if energy for the hydroponics was sourced only from a renewable source such as solar, the environmental impacts of the hydroponically produced lettuce would have been lower compared to conventional agriculture [33].

## 7. Challenges and Future of Indoor Production

Although indoor vertical farming has been advocated for some time (beginning in 1999), only a handful of commercially viable vertical farms have been established since that time [90]. These include establishments like Rural Development Authority (Sejong City, South Korea), Plant factories (numerous, 50+), Nuvege (Kyoto, Japan), Sky Green (Singapore), Alterrus (Vancouver, Canada), The Plant (Chicago, IL, USA) [90], Aero Farms (Newark, NJ, USA), Vertical Harvest Farm (Jackson, WY, USA) [15]. Still, the concept of IF within city limits is still too new to make an assertion that this technology driven production systems will be successful on a worldwide scale from both an economic and social point of view. Furthermore, it will take many years for their impact on ecological processes to be manifested in terms of global climate change. However, some early life cycle assessments indicate the impacts of IF on climate change may depend on the location of the facility mainly because of the current higher electricity requirement compared to conventional agriculture as well as the source of the electricity used [33].

Economic challenges such as bankruptcy pose a big test for the burgeoning IF industry. Several vertical indoor farms including FarmedHere in Bedford Park, IL, USA, Potponics in Georgia, USA and a host of others have suffered this fate [12]. Nevertheless, others such as Urban Produce and Plenty (San Francisco, CA, USA), Plantagon (Stockholm, Sweden) and Aerofarm (Newark, NJ, USA) are operating with a margin of profit although this information is not readily available [12]. Aerofarm operates from approximately a 6503 m^2^ (70,000 ft^2^) facility, which is among the largest producing facility in the United States and possibly the whole world [12]. Other large production units are operating in countries such as Japan, Singapore, South Korea and Taiwan. Most likely, some or all of them are receiving direct or indirect government subsidies, which has contributed to their sustenance [189]. A wholistic analysis is needed to ascertain why some of the companies have gone bankrupt, whereas others seem to be making profit. Though a great innovation, it is probable that larger scale IFs can outcompete smallholder farmers. This is because small holder farmers in low-income countries who operate small fruit and vegetable farms often lack extensive government support and are faced with post-harvest losses and poor functioning supply and value chains.

Vegetable production in IFs such as greenhouses, grow tents, and indoor vertical farms requires artificial light, with or without addition of natural light for plants to undergo photosynthesis. Artificial lights within these structures are often provided by LED lights which require electricity [15] making energy a major cost. Since majority of modern greenhouses and IFs are located in close proximity to cities, critics believe that they are going to compete with the already stretched residential electricity supply. Therefore, regular supply of electricity and the cost of electricity can pose as a production challenge. If developed countries such as United States adopt vertical farms extensively, the country’s energy need can increase up to eight times from the current amount generated by all the power plants [18]. Erecting solar panels in cities may not be the solution to supply all the electricity needs since cities are shaded by buildings and lack light penetration of the natural environment [15,18]. The invention of programmable and more efficient LEDs with the ability to switch on and off when needed can help reduce consumption and lead to energy cost savings. Similarly, strides being made on the use of pulsed lighting could also help mitigate energy costs associated with indoor production [190]. Also, advances being made in the use of AI and machine learning for optimizing growing conditions in IFs could also lead to enhanced efficiency [33]. Other technological advancements that are likely to reduce the energy costs of IF include plant breeding and genetic engineering, co-location, co-generation, and symbiotic systems and expanding renewable energy sources for use [33]. IF specific crops could be developed to have attributes such as uniform and early fruiting, rapid biomass and multi-harvest capability, desirable architecture for auto-harvesting and quality improvement (i.e., color or flavor) in response to LEDs. These can either reduce the energy cost of harvesting or increase yield which can in turn be used to offset energy cost and use [33]. Co-location, and co-generation systems involve placing farms next to abandoned or underutilized assets and taking advantage of them. For example, Great Northern Hydroponics in Quebec Canada is located by a power plant and uses a cogeneration machine to capture excess heat from the power plant and recycles it into their greenhouse hydroponic operation thereby reducing reliance on fossil fossils and reducing their heating costs as well [33]. Symbiotic systems are anticipated to integrate with municipal infrastructure including heating, biogas, waste, water, and energy with food production. The goal is to create high efficiency growing systems by recycling and using waste generated from municipalities [33]. Expanded use of renewable energy including combining solar with wind and thermal energy could reduce energy costs. Similarly, exploring fiber technology in terms of directing outdoor natural lights indoors will be another potentially energy-efficient option [33].

Consumer acceptance of vegetables produced under controlled environments could affect the future of these production systems. It is important to know if communities are willing to accept such foods since they are produced with less water, and no pesticides, or they will reject them because they are produced using unconventional methods. Under some jurisdictions like the United States (US) and European Union (EU), will they conform to the definition of organic foods since they are produced without soil? There is limited information that targets the assessment of consumer reaction to IF produce. Coyle and Ellison [191] reported that consumers did not find any difference among lettuce produced in vertical indoor facilities, greenhouses or in the open field. However, they expressed their hesitation about the unconventional production method. Notwithstanding, relative price may still be the determinant factor for consumer demand. If consumers are to pay premium prices for vegetables produced in indoor farms, low-income consumers may miss out on the nutritional benefits.

Advocacy groups could emerge in opposition to foods produced in indoor farms. They may be similar to groups that have resisted genetically engineered foods. Cox [192] presented a dissenting view about IF, with regards to the use of artificial light. His argument is that it does not make sense to grow plants under artificial light, however, the author failed to address the potential nutritional benefits. Even so, he does not oppose forms of IF in which plants receive their energy directly from sunlight. Some people also believe that massive expansion of IFs can take away the market for smallholder vegetable producers [12]. However, there will be a market large enough to accommodate both indoor and outdoor producers because inaccessibility of vegetables is widespread in many urban areas leading to an enormous problem of micronutrient deficiency which needs to be addressed. Avgoustaki and Xydis [193] perceive indoor urban vertical farming as a new foundation in the urban food production system which is opening doors to other sustainable activities including energy and grey water recycling, creativity and skills development on sustainable food production to feed urbanites with fresh and nutritious produce. A fast-growing population coupled with the fact that more people live in urban areas worldwide (i.e., urban expansion), reveals the importance of seeing indoor production as a necessary complimentary production system to outdoor productions systems and not as a competitor. This is because synergy will be required to meet the challenge of providing adequate food for the projected 9.7 billion people by 2050.

Typically, most large high-tech greenhouse operations are located several miles away from most urban centers [90]. This is because land becomes cheaper the further you move away from the city. Hence most commercial growers harvest their produce before they are fully ripened, particularly with delicate fruits and vegetables such as tomatoes and strawberries [90]. This is seen as a necessity by the industry and allows them to ship their produce over long distances without major damage from handling and packaging. However, moving greenhouses close to cities will reduce or totally eliminate this drawback, and will add the option of buying locally grown, safe and ultra-fresh (i.e., hours old) produce on demand. Future evolution of greenhouses within cities can be targeted at stacking high tech greenhouses on top of one another, thereby reducing their architectural footprint. Establishing IFs within cities can lead to the creation of several job opportunities ranging from greenhouse management, information technology management, human resources management to outreach and community education. Hence, establishing IF in urban settings can create low to high tech employment prospects for people.

Improving the economic, social, and environmental aspects and sustainability of IF requires continued and increased investment in research and the development of good supportive policies and strategies. Increased funding can spur further innovations in LED light technologies with respect to efficiency, distribution, improvement of produce quality and yield maximization. For example, further research can be done to develop more efficient smart hybrid light systems which combine daylighting systems, dimmable LEDs, and internet-of-things light sensors to improve automatic real time management of PPFD levels needed for optimal plant growth in IF [27]. Similarly, current LED light technologies developed for reducing shading in IF facilities (i.e., GreenPower LED interlighting module by Phillips) and those created for improved LED color mixing and closer placement to crops (i.e., CoolGrow by MechaTronix) can be further optimized. In addition, newer technologies can be developed to generate suitable light recipes for meeting the lighting needs for varied crop plants [27] and for turning out produce with optimized organoleptic properties. More so, since there is growing interest in precision and personalized nutrition [194], light recipes can be developed for growing produce very rich in certain target nutrients like iron to meet personalized diet needs of people. Also, light redirection technologies including optical fiber daylighting systems can be further researched and improved for enhancing uniform light coverage in IF facilities. These developments can ultimately boost productivity in IF facilities because stacking for example in vertical farms often limits light penetration and uniform distribution on the lower canopy of crops [27,195]. This hampers overall optimal growth and yield. More research is also needed for improving other engineering controls used in IF including cooling and ventilation technologies as well as water and nutrient management systems [26,27,31] to make them more efficient.

Robust models are needed for analyzing and verifying the energy performance of IF facilities [24]. Such models can also be used to guide the development of advanced and more energy efficient IF facilities. In the same vein, quantitative research can be conducted to assess the impacts of IF on land use and greenhouse emissions, non-point source pollution, land degradation relative to open field cultivation. Similarly, it would be important to assess differences in the carbon footprint of IF food production in urban areas compared to moving food over hundreds or thousands of miles to outlets in cities [196]. Ultimately accurately quantifying both the advantages and disadvantages of all types of IF will be useful for their broader acceptance since their appeal is currently not generally positive [197,198,199]. Since the impacts of climate change include dwindling freshwater resources and expansion of dryland areas, it will be important to examine how IF could impact food production in dry regions and environments with extreme conditions [26,196].

With regards to the plants grown in IF, there is a need for developing technologies for rapid detection and analysis of disease and physiological stresses [27,31]. Such systems can enhance rapid detection of potential problems and will ultimately reduce plant loss and boost yield. Plant growth promoting rhizobacteria (PGPR) are generally believed to enhance plant growth and resilience in both soil and soilless systems [35]. As such, research can be done to further explore their use as an amendment in IF production systems to enhance plant growth, nutrient uptake, and resistance to biotic and abiotic stresses. Their use could minimize the use of chemical fertilizers and other crop care products in IF [200,201]. Research efforts like this could lead to IF specific microbiome engineering. Computing tools can also be developed for modeling plant growth in IF environments and facilitate the ability of farmers to grow produce with targeted nutrition.

Since crop production using IF is still quite expensive, long term research goals in this area should include how to make IF equipment cheaper so that people living in low- and middle-income countries can afford them and benefit from the technology [32]. Along this line, a good policy framework should be formulated on standardizing the technology and practices of the industry. Though strides are being made in Europe, IF currently lack certification programs in most parts of the world. The Association of Vertical Farms and Control Union UK plan to launch a certification program for the industry this year (i.e., 2022) in the UK [202]. Creating sustainability standards for the IF industry will permit efficiency and output analyses among farms and facilitate the sharing of ideas across the industry [203]. Like most food and food production standards and regulations, an IF certification standard must be designed with food safety, health, and social justice in mind.

## 8. Conclusions

Since technological options for reducing the impacts of agriculture on the environment are limited amid the dire need to increase food production to sustain an exponentially growing human population, the new opportunities provided by IF should be explored extensively to enhance food production to meet our growing needs in a way that is in balance with maintaining a sound environment. IF has the potential to fill production gaps that currently exist and offer great opportunities for growing and eating safe locally grown food especially in this era of climate change.

As a relatively young industry, investments in research are needed to address the current challenges it faces including efficiency and sustainability of IF facilities and growing systems. Formulating good policies to accompany the scientific innovations will create a resilient IF industry that can sustainably produce food to supplement outputs from traditional farming activities. By so doing, IF will strengthen produce supply chains to meet our food needs.

## Figures and Tables

**Figure 1 plants-11-02843-f001:**
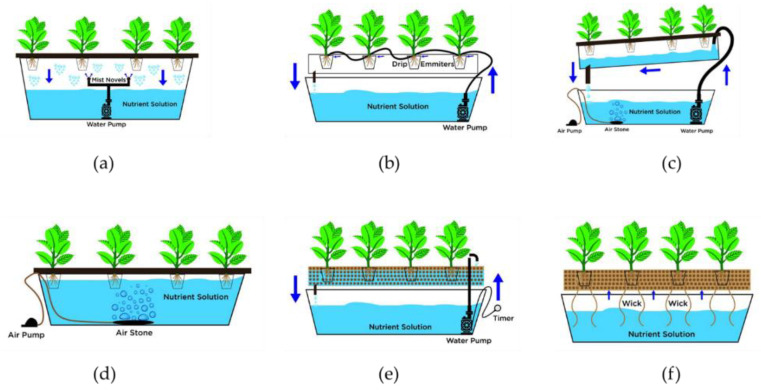
Common hydroponic systems: (**a**) aeroponics, (**b**) drip system, (**c**) nutrient film technique (NFT), (**d**) deep water culture, (**e**) ebb and flow and (**f**) wick system. The illustrations are sourced from The Hydroponics Guru. https://thehydroponicsguru.com/types-of-hydroponics-systems/ (accessed 12 October 2022) [97].

**Figure 2 plants-11-02843-f002:**
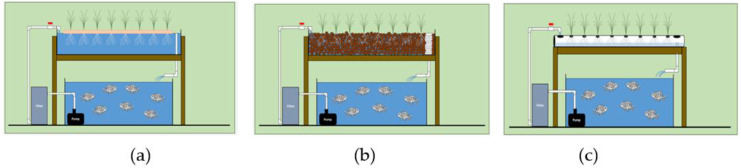
Popular aquaponics systems: (**a**) deep water culture (DWC), (**b**) ebb and flow system, (**c**) nutrient film technique system. These diagrams are sourced from [103].

**Table 1 plants-11-02843-t001:** List of some leafy vegetables and herbs studied for indoors production ^†^.

Vegetable Crop	Scientific Name	References
^‡^ Amaranth	*Amaranthus* spp.	Ebert et al. [52]; Kyriacou et al. [53]; Ampim et al. [54]; Rocchetti et al. [55]
^‡^ Arugula/Garden Rocket	*Eruca sativa* L.	Murphy and Pill [56]; Berba and Uchanski [57]; Wuang et al. [58]; Ying et al. [59]; Pennisi et al. [60]; Pennisi et al. [61]
^‡^ Basil	*Ocimum basilicum*	Chandra et al. [62]; Piovene et al. [63]; Pennisi et al. [60]; Pennisi et al. [61]
Bayam Red	*Amaranthus gangeticus*	Wuang et al. [58]
^‡^ Broccoli	*Brassica oleracea* var. italica	Kyriacou et al. [53]; Wuang et al. [58]; Sun et al. [64]
^‡^ Broccoli/^‡^ Choy sum/^‡^ Pac Choi/Pak Choy	*Brassica rapa* subsp. *chinensis* var. parachinensis	Borrelli et al. [65]; Wuang et al. [58]; Niu et al. [66]; Niu et al. [67]; Kyriacou et al. [53]
Chinese Broccoli	*Brassica alboglabra*	He et al. [68]
^‡^ Cabbage	*Brassica oleracea* var. capitata	Ying et al. [59]
^‡^ Chicory/ Catalogna	*Cichorium intybus*	Maucieri et al. [69]; Pennisi et al. [60]; Pennisi et al. [61]
^‡^ Cilantro/Coriander/dhanial	*Coriandrum sativum*	Kyriacou et al. [53]
^‡^ Egyptian spinach	*Corchorus olitorius*	Ampim et al. [54]
^‡^ Ice plants	*Mesembryanthem crystallinum*	He et al. [68]
^‡^ Kale/Red Kale Kale “Red Russian”	*Brassica oleracea* *Brassica napus* L.	Chandra et al. [62] Ying et al. [59]
^‡^ Lettuce	*Lactuca sativa*	Borrelli et al. [65]; Ngilah et al. [70]; Pinto et al. [71]; Kyriacou et al. [53]; Niu et al. [66]; He et al. [68]; Loconsole et al. [72]; Maucieri et al. [69]; Gómez and Jiménez [73]; Pennisi et al. [60]; Pennisi et al. [61]; Su et al. [74]
^‡^ Malabar spinach/ Vine spinach	*Basella alba*	Muchjajib et al. [75]
^‡^ Mustard	*Brassica juncea* L.	Muchjajib et al. [75]; Ying et al. [59]
^‡^ Parsley	*Petroselinum crispum*	Chandra et al. [62]
^‡^ Radish/Rat-tail radish	*Raphanus sativus* L.	Berba and Uchanski [57]; Muchjajib et al. [75]; Kyriacou et al. [53]
^‡^ Red beet	*Beta vulgaris*	Kyriacou et al. [53]
^‡^ Red cabbage	Cabbage (*B. oleracea* var. capitata), Red and purplemustard (*B. juncea* Czem.), mizuna (*B. rapa* L. var. nipposinica), and purple kohlrabi (*B. oleracea* L. var. gongylodes L.)	Kyriacou et al. [53]; Berba and Uchanski [57]
^‡^ Spinach	*Spinacia oleracea*	Borrelli et al. [65]; Saaid et al. [76]
^‡^ Swiss Chard/Chard/Table beet	*Beta vulgaris* subsp. *vulgaris* var. vulgaris	Chandra et al. [62]; Maucieri et al. [69]; Murphy et al. [77]
Water spinach/Kangkong	*Ipomoea aquatica*	Muchjajib et al. [75]

^†^ These references cover between 2010 and 2020. ^‡^ Leafy vegetable crop grown as both microgreens and matured crops.

**Table 2 plants-11-02843-t002:** Facilities investigated for producing leafy vegetables indoor ^†^.

Type of Infrastructure	References ^†^
Air-conditioned laboratory	Ngilah et al. [70]; Berba and Uchanski [57]
Greenhouse	Murphy and Pill [56]; Murphy et al. [77]; Ebert et al. [52]; Pinto et al. [71]; Kyriacou et al. [53]; Niu et al. [66]; Maucieri et al. [69]
Growth chamber/Indoor	Piovene et al. [63]; Sun et al. [64]; Kyriacou et al. [53]; Niu et al. [66]; Loconsole et al. [72]; Penissi et al. [60]; Gómez and Jiménez [73]; Niu et al. [67]; Pennisi et al. [61]; Ying et al. [59]
Grow tent (high tunnel)	Borrelli et al. [65]
Rooftop farming	Su et al. [74]

^†^ These references cover between 2010 and 2020.

**Table 3 plants-11-02843-t003:** Grow media or systems examined for indoor leafy vegetable production ^†^.

Grow Medium/System	References ^†^
Aeroponics	Chandra et al. [62]
Hydroponics	Murphy et al. [77]; Saaid et al. [76]; Kyriacou et al. [53]; Niu et al. [66]; Loconsole et al. [72]; Maucieri et al. [69]; Pennisi et al. [60]; Gómez and Jiménez [73]; Pennisi et al. [61] Pennisi et al. [91]; Su et al. [74]
Blends/mixes:	
Peat moss and vermiculite	Ebert et al. [52]; Kyriacou et al. [53]; Pennisi et al. [91]
Compost, peat, coir, and perlite	Ying et al. [59]
Coconut coir dust, peat	Muchjajib et al. [75]
Coconut coir dust, sugarcane filter cake	Muchjajib et al. [75]
Coconut coir dust, vermicompost	Muchjajib et al. [75]
Coconut coir dust	Muchjajib et al. [75]
Garden soil/Potting soil	Wuang et al. [58]; Niu et al. [67]
Jute and kenaf fibers	Di Gioia et al. [92]
Mats (Sure to Grow) consisting of polyethyleneterephthalate	Di Gioia et al. [92]
Mats (Sure to Grow) consisting of polyethyleneterephthalate	Berba and Uchanski [57]
Paper towel/pad	Murphy et al. [77]; Ebert et al. [52]; Sun et al. [64]
Peat moss/Peat-lite	Murphy et al. [77]; Murphy and Pill [56]; Muchjajib et al. [75]; Di Gioia et al. [92]
Rock wool	Loconsole et al. [72]
Sand	Muchjajib et al. [75]
Sugarcane filter cake	Muchjajib et al. [75]
Textile fiber mat	Di Gioia et al. [92]
Vermicompost	Muchjajib et al. [75]
Vermiculite	Murphy and Pill [56]; Murphy et al. [77]
Volcanic growing media	Piovene et al. [63]

^†^ These references cover between 2010 and 2020.

**Table 4 plants-11-02843-t004:** Aquaponics system components: relevance, considerations for selection and examples.

Component	Importance	Considerations	Example(s)
Fish tank	Fish culture	1. The size and shape should be suitable for accommodating fish without stressing them 2. Tank condition and history (new or used) 3. Material used to manufacture the tank (e.g., polypropylene, high-density polyethylene, fiberglass, PVC or EPDM lining material, low-density polyethylene (LDPE)) 4. Budget 5. Color: light color preferred for practical reasons including viewing and reflection of sunlight for temperature moderation 6. Resistance to UV 7. Failsafe and redundancy	Liquid totes, animal watering tanks
Fish tank cover	1. Prevent growth of algae 2. Prevent fish from jumping out	Suitability and durability	1. Cloth, 2. tarps, 3. woven palm fronds, 4. plastic lids, 5. shading nets
Filtration system (mechanical or passive and biological)	1. Mechanical solid removal 2. Conversion of ammonia from fish waste to nitrite and then nitrate for plant use thus maintaining the overall chemical steadiness of the aquaponic system	1. Use a productive biofilter that maintains adequate levels of dissolved oxygen to support nitrification	1. Solid separators: swirl (vortex), clarifier, radial flow, solid filters (mechanical), raft filters, bird netting, screen filters, filter socks 2. Biofilters: moving bed filters, static filters and drip
Water flow and aeration	1. Conveyance of wastes to filters and nutrient-rich water to plants (water flow) 2. Maintaining sufficient dissolved oxygen in the system to support the survival of fish, beneficial microbes and plant growth	Use proper size water and air pumps	1. Water pump: small or large submersible or inline 2. Aeration: active aqua air pumps, alita linear diaphragm air pumps, cylinder air stone, air bubble diffuser
Fish Type	Provide nutrients for plant growth	1. adaptability, 2. resilience, 3. diets and 4. breeding habits	1. Tilapia, 2. ornamental fish, 3. cat fish, 4. perch, 5. koi, 6. gold fish, 7. tropical fish
Plants	Use nutrients from the system and therefore act as a natural filter	1. suitability, 2. ease of cultivation	1. Lettuce, 2. watercress, 3. basil, 4. tomatoes, 5. peppers, 6. cucumbers, 7. cauliflower, 8. strawberries

**Table 5 plants-11-02843-t005:** Merits and demerits of the various hydroponic systems.

Hydroponic System	Merits	Demerits
Aeroponics	Superior availability of oxygen to roots Economical in terms of water and nutrient use Provides higher growth rate Small space requirement Easy to move around	It can be costly to set up Requires high levels of knowledge and skill to manage Needs to be monitored all the time Needs constant power supply Technical system problems can lead to plant death and financial loss
Drip	Relatively cheap to set up It is flexible and scalable It is low maintenance compared to other systems It is less likely to fail	This system can be complicated for small-scale operations. Similarly, maintenance can be high if water is recycled Use of non-recovery designs can be wasteful
Nutrient Film Technique	Can be easily established Relatively low construction cost Availability of oxygen to plant roots. It is a low-waste system	Dense root mass can impede nutrient solution flow through the systems Disease can be easily spread Root death can be a problem
Deep Water Culture	Faster plant growth because of superior nutrient and oxygen uptake Maintenance needs are little Easy to assemble since there are few moving parts Nutrient solution top up is easy	Air pump disfunction can impair aeration and affect plant growth Temperature moderation can be a challenge in a non-recirculatory system System cleanup requires taking it out of operation
Ebb and Flow	Its construction cost is low Plants can easily access nutrients sufficiently The system is easy to use	High potential for spread of root diseases Water and nutrient use are inefficient Growing medium needs periodic replacement Managing pH of the system can be challenging Oxygen supply may be limited in the ebb or drain stage Plants that grow extensive root systems have a potential for tangling and this can be problematic during harvesting Breakdowns can occur often
Wick	It is easy to build and maintain Presents opportunities for repurposing household items and materials It uses less water and nutrients It does not require the use of electricity	It is not suited for cultivating large and fruit bearing plants High nutrient build up in the growing medium is possible overtime Periodic monitoring is needed Uneven distribution of water and nutrients is possible

**Table 6 plants-11-02843-t006:** Scales of use of hydroponic and aquaponic growing systems.

Growing System	Scale of Use
*Hydroponic*	
Aeroponics	Domestic and commercial
Drip	Domestic and commercial
Nutrient Film Technique	Domestic and commercial; most scalable hydroponic technique and one of the most adopted methods
Deep Water Culture	Domestic and commercial; good starting techniques to explore for beginners
Ebb and Flow	Domestic and commercial; first commercial hydroponic system
Wick	Mostly domestic and not commercial
*Aquaponics*	
Nutrient Film Technique	Domestic and commercial, mostly used for lettuce production
Deep Water Culture	Domestic and large-scale commercial
Ebb and Flow	Domestic and commercial

**Table 7 plants-11-02843-t007:** The research focus of plant growth factors and parameters relevant to indoor vegetable production ^†^.

Research Focus	References ^†^
Factors affecting growth: Light, photoperiod, water, temperature	Berba and Uchanski [57]; Borrelli et al. [65]; Saaid et al. [76]; Ngilah et al. [70]; Piovene et al. [63]; Wuang et al. [58]; Niu et al. [66]; He et al. [68]; Loconsole et al. [72]; Pennisi et al. [60]; Gómez and Jiménez [73]; Pennisi et al. [61]; Niu et al. [67]; Ying et al. [59]
Growth parameters: Crop yield, plant size, leaf color	Murphy and Pill [56]; Murphy et al. [77]; Borrelli et al. [65]; Chandra et al. [62]; Ebert et al. [52]; Piovene et al. [63]; Ngilah et al. [70]; Wuang et al. [58]; Niu et al. [66]; He et al. [68]; Loconsole et al. [72]; Maucieri et al. (2019); Pennisi et al. [61]; Niu et al. [67]; Ying et al. [59]
Phytochemical composition/nutrients	Chandra et al. [62]; Ebert et al. [52]; Muchjajib et al. (2014); Pinto et al. [71]; Sun et al. [64]; Piovene et al. [63]; Kyriacou et al. [53]; Niu et al. [66]; Rocchetti et al. [55]; Niu et al. [67]

^†^ These references cover between 2010 and 2020.

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
