# Peer review of "Indoor Vegetable Production: An Alternative Approach to Increasing Cultivation"

_plants, 2022, doi:10.3390/plants11212843_

Round 1

Reviewer 1 Report

The manuscript entitled “Indoor Vegetable Production: An Alternative Approach to In- creasing Cultivation” which was submitted in Plants. My suggestions/recommendations are as follow:

You have mentioned about the vegetable production under greenhouses, indoor farms, high tunnels, and screenhouses and can control of production factors such as temperature, relative humidity, and carbon dioxide, as well as extension of the growing season. But you did not mention the novelty statement of this review. What you have done something novel in your review?

Reduce the number of keywords to 5 and should not repeat the words which you already mention in the title.

“However, the current projections indicate the level of food production success attained thus far may not continue 2050 and beyond”. Need reference.

“This is because it provides growers the ability to create the desired conditions for crop growth regardless of outdoor weather conditions” what are some other factors which affects the crop productivity when we grown it in the field environment?

“Therefore, this review paper discusses indoor production of primarily leafy vegetables including the growing systems and types of facilities used, factors affecting production, plant nutrition, economics, challenges and the future prospects of indoor farming”. These objectives are not enough, please also mention which idea were in your mind which encourages you to write this review? What is the main novelty in this manuscript? Please write all these points and re-write the objectives again.

In the greenhouse portion, you did not mention the advantages to grow over hydroponic environment in the greenhouse? Please mention something about it.

“Studies show that applying supplemental light of various qualities (UV-A, blue, green, red, far-red, and white LED) had significant effect on phytochemicals content (anthocyanins, carotenoids, chlorophylls and flavonoids) of lettuce leaves” grammar mistake.

  “Magnesium, Iron, Potassium, Zinc, and Phosphorus, among others” use the abbreviation.

“Photosynthesis is driven mainly by red and blue light consequently providing the right doses of these lights can efficiently promote plant growth.” Need latest citation.

And also mention some more future recommendations from your review.

Author Response

Comment 1: You have mentioned about the vegetable production under greenhouses, indoor farms, high tunnels, and screenhouses and can control of production factors such as temperature, relative humidity, and carbon dioxide, as well as extension of the growing season. But you did not mention the novelty statement of this review. What have you done something novel in your review?

“Therefore, this review paper discusses indoor production of primarily leafy vegetables including the growing systems and types of facilities used, factors affecting production, plant nutrition, economics, challenges and the future prospects of indoor farming”. These objectives are not enough, please also mention which idea were in your mind which encourages you to write this review? What is the main novelty in this manuscript? Please write all these points and re-write the objectives again.

Response: These comments have been addressed. Please refer to lines 113-131 in the revised manuscript.

 Comment 2: Reduce the number of keywords to 5 and should not repeat the words which you already mention in the title.

Response: the number of keywords has been reduced to 5. Please see lines 24-25 in the revised document.

 Comment 3: “However, the current projections indicate the level of food production success attained thus far may not continue 2050 and beyond”. Need reference.

Response: This comment was concocted based on projections for population growth and food production found in the literature and not taken from any journal article.

 Comment 4: “This is because it provides growers the ability to create the desired conditions for crop growth regardless of outdoor weather conditions” what are some other factors which affects the crop productivity when we grown it in the field environment?

Response: Soil factors including fertility, pathogens, pests, salinity as well as weeds and topography have been added to the discussion. Please see lines 48-53 in the revised document.

 Comment 5: In the greenhouse portion, you did not mention the advantages to grow over hydroponic environment in the greenhouse? Please mention something about it.

Response: The manuscript was designed to address facilities used separately from growing systems/techniques. However, following your comment, we added a sentence on the advantage of doing hydroponic production in a greenhouse. Please refer to lines 144-145 in the revised document.

Comment 6: “Studies show that applying supplemental light of various qualities (UV-A, blue, green, red, far-red, and white LED) had significant effect on phytochemicals content (anthocyanins, carotenoids, chlorophylls and flavonoids) of lettuce leaves” grammar mistake.

Response: The grammatical mistake has been corrected. Please see lines 426-427.

 Comment 7: “Magnesium, Iron, Potassium, Zinc, and Phosphorus, among others” use the abbreviation.

Response: The names were replaced by symbols as suggested. Please see line 446.

Comment 8: “Photosynthesis is driven mainly by red and blue light consequently providing the right doses of these lights can efficiently promote plant growth.” Need latest citation.

Response: A 2022 citation has been provided. Please refer to line 443 in the revised manuscript.

Comment 9: And also mention some more future recommendations from your review.

Response: Additional recommendations have been provided. Please refer to lines 795-851.

Reviewer 2 Report

The presented manuscript is an interesting work, containing a lot of information. The subject matter chosen by the authors is very extensive, which can be seen from the volume of this work. Such a broad approach to the topic gives a general approach, however, it causes the topics to be discussed generally, not in detail.

Before publishing it, I suggest that you supplement this article with some information. The article lacks information on the discussed systems, i.e. which of these systems are used in practice and on what scale.

I also suggest that the authors read the work again in order to remove minor errors, e.g. punctuation.

Author Response

Comment 1. The presented manuscript is an interesting work, containing a lot of information. The subject matter chosen by the authors is very extensive, which can be seen from the volume of this work. Such a broad approach to the topic gives a general approach, however, it causes the topics to be discussed generally, not in detail. Before publishing it, I suggest that you supplement this article with some information. The article lacks information on the discussed systems, i.e. which of these systems are used in practice and on what scale.

Comment 2. I also suggest that the authors read the work again in order to remove minor errors, e.g. punctuation.

Response to both comments: We read over the manuscript again and corrected the minor errors we found.  To your suggestion, we provided information on the scale of use of the growing systems. Please see lines 410-418.

Round 2

Reviewer 1 Report

The authors reviewed “Indoor Vegetable Production: An Alternative Approach to In-creasing Cultivation”, More examples are provided for providing more adequate food for humans. The topic selection of the review is of great significance, and the article needs to be revised before it is published.

1.Abstract.This paper reviews these alternative vegetable production approaches to reveal the need for exploring them to increase crop production. Ask the author to add some examples of specific planting patterns.

2.line 151-152. First, they are scientifically designed to increase light exposure, save cost, and reduce 151 environmental impact through the use of biodegradable plastic films. This is very interesting. Can you give me a little more explanation? Biodegradable plastic control-insistent pang is through what way to increase the light exposure?

3.Table 1. The table can also be optimized to make the second column wider.

4.Line 614 when K was increased from 0 to 8 mM K in nutrient solution.I think it should be when K was increased from 0 to 8 mML-1 K in nutrient solution.

5. This paper reviews in detail the advantages and disadvantages of facility cultivation for alleviating food shortage, especially vegetable production, but it is too lengthy and can be appropriately brief.

Author Response

Please send the attached cover letter for our responses.  Thank you.
